# Iron Overload in Children with Acute Lymphoblastic and Acute Myeloblastic Leukemia—Experience of One Center

**DOI:** 10.3390/cancers16020367

**Published:** 2024-01-15

**Authors:** Malgorzata Sawicka-Zukowska, Anna Kretowska-Grunwald, Agnieszka Kania, Magdalena Topczewska, Hubert Niewinski, Marcin Bany, Kamil Grubczak, Maryna Krawczuk-Rybak

**Affiliations:** 1Department of Pediatric Oncology and Hematology, Medical University of Bialystok, Jerzego Waszyngtona 17, 15-274 Bialystok, Poland; annammkretowska@gmail.com (A.K.-G.); agnieszka.kania@umb.edu.pl (A.K.); hubniewinski@wp.pl (H.N.); marcinbany9@gmail.pl (M.B.); rybak@umb.edu.pl (M.K.-R.); 2Faculty of Computer Science, Bialystok University of Technology, Wiejska 45A, 15-351 Bialystok, Poland; m.topczewska@pb.edu.pl; 3Department of Regenerative Medicine and Immune Regulation, Medical University of Bialystok, Jerzego Waszyngtona 13, 15-269 Bialystok, Poland; kamil.grubczak@umb.edu.pl

**Keywords:** ferritin, iron overload, childhood leukemia, acute lymphoblastic leukemia, acute myeloblastic leukemia, blood transfusion

## Abstract

**Simple Summary:**

Post-transfusion iron overload is a common side effect of anticancer treatment, including leukemias. We showed that serum ferritin levels could be valuable prognostic marker of death in children with AML and ALL. Importantly, improved survival of leukemia patients was found to be associated with lower ferritin values before therapy. It is noteworthy that the observed phenomenon was dependent on the cancer type, sex, and age of the studied patients.

**Abstract:**

Transfusions of packed red blood cells (PRBCs), given due to an oncological disease and its acute complications, are an indispensable part of anticancer therapy. However, they can lead to post-transfusion iron overload. The study aim was to evaluate the role of ferritin as a nonspecific marker of leukemic growth and marker of transfusion-related iron overload. We performed a longitudinal study of PRBC transfusions and changes in ferritin concentrations during the oncological treatment of 135 patients with childhood acute lymphoblastic and acute myeloblastic leukemia (ALL and AML, median age 5.62 years). At the diagnosis, 41% of patients had a ferritin level over 500 ng/mL, and 14% of patients had a ferritin level over 1000 ng/mL. At the cessation of the treatment, 80% of the children had serum ferritin (SF) over 500 ng/mL, and 31% had SF over 1000 ng/mL. There was no significant difference between SF at the beginning of the treatment between ALL and AML patients, but children with AML finished treatment with statistically higher SF. AML patients had also statistically higher number of transfusions. We found statistically significant positive correlations between ferritin and age, and weight and units of transfused blood. Serum ferritin at the moment of diagnosis can be a useful marker of leukemic growth, but high levels of SF are connected with iron overload in both AML and ALL.

## 1. Introduction

Ferritin, a protein involved in the metabolism of iron, was first described in the literature by a German pharmacologist, Oswald Schmiederberg, who found the presence of an iron-rich component in a liver of a horse. For over 70 years, ferritin has been analyzed for its numerous functions, including its roles in iron metabolism, angiogenesis, inflammation, immunosuppression, and proliferation. Although its intracellular functions are quite thoroughly described, the function of extracellular form (plasma) of ferritin is still not completely clear [1].

Although only a small percentage of ferritin supplies are found in the blood serum and the majority remains hidden intracellularly, the serum ferritin concentration reflects adequately its storage pool—measured in plasma, it provides fast and reproducible information about iron supplies in the body [2,3]. Studies describing the changes in ferritin concentrations monitored over the years after chemotherapy confirmed the long-term persistence of its elevated serum concentration [4,5]. High ferritin plasma levels evaluated prior to treatment have been reported among patients with a variety of cancers, these included leukemia, Hodgkin’s lymphoma, neuroblastoma, kidney cancer, liver cancer, melanoma, non-small cell lung cancer, pancreatic cancer, and breast cancer [1,6,7]. A strong correlation has been demonstrated between high ferritin levels and advanced disease stage, worse treatment response, and poor prognosis. High ferritin levels before the bone marrow transplant surgery among patients with proliferative bone marrow diseases seem to be linked to worse prognosis, when compared to those with low ferritin levels [8]. 

In recent years, therapeutic protocols of pediatric neoplasms have been tightened, which led to the necessity of more frequent transfusions of packed red blood cell concentrate (PRBCs) in children undergoing oncological treatment. Transfusions of blood products and components allow for the maintenance of treatment of the underlying disease. In this group of patients, anemia significantly worsens the effectiveness and quality of treatment, and transfusions of blood and its components have greatly contributed to the increase in survival and tolerance of treatment [9]. The most common side effect, connected with multiple blood transfusions, is iron overload (IO), routinely evaluated after procedures. 

Exogenous iron acquired from transfusion can accumulate rapidly. Data from studies on patients with chronic hemolytic anemias who require regular blood transfusions clearly show that just 10–20 transfusions may already reveal laboratory and histological signs of iron overload [10,11,12,13]. Each full unit of PRBCs provides the body with approximately 200–250 mg of iron [14]. It was also shown that the frequency of transfusion in the amount of 2 units per month used in non-malignant hematological disorders will result in the accumulation of 20 g of extra iron in the body within 4 years [15]. Since there is no physiological mechanism to clear out iron excess, regular and repeatable transfusions lead to rapid accumulation of iron in macrophages of the reticuloendothelial system, which in turn leads to the deposition of excess iron in hepatocytes, myocardium, and endocrine organs, resulting in a cytotoxic effect among free radicals [16,17]. The most frequently observed symptoms associated with IO include thyroid dysfunction, growth hormone deficiency, diabetes, and delayed puberty [18]. However, the most dangerous outcomes for patients are heart and liver failure. The main cause of morbidity and mortality in the group of patients with transfusion-dependent thalassemia is iron-dependent cardiomyopathy [19,20].

It is estimated that as many as 97% of children treated for acute lymphoblastic leukemia (ALL) require transfusions of red blood cells [21]. The interaction between iron overload, chemotherapy, and radiotherapy has not been well studied; however, organ toxicity of these three phenomena overlaps, which may lead to exacerbation of early and late complications of chemotherapy and radiotherapy [11]. Those who undergo the procedure of hematopoietic stem cell transplantation have the highest levels of serum ferritin as a consequence of higher number of transfusions (Ref. [22]), and are at increased risk of early mortality post-transplantation [8]. IO can be measured indirectly with serum ferritin concentration, regarded as the most accessible and cheapest indicator of IO. Identifying the group of pediatric oncological patients endangered with iron overload and its consequences is important in minimizing the long-term morbidity and increase quality of life among survivors of childhood cancer [23,24]. The aim of the study was to evaluate the changes in ferritin concentration due to transfusion during antineoplastic treatment. In addition, we analyzed the prognostic value of iron levels in an assessment of death risk among children with acute leukemias. 

## 2. Materials and Methods

### 2.1. Patients

The study population was selected from patients treated for acute lymphoblastic leukemia (ALL) and acute myeloblastic leukemia (AML) who received blood transfusions in the Department of Pediatrics, Oncology, and Hematology in the Medical University of Bialystok, Poland. The study group consisted of 551 children (only these patients, who required blood transfusion). The study was conducted retrospectively. Inclusion criteria were as follows: age 0–18 years; diagnosis with acute lymphoblastic leukemia or acute myeloblastic leukemia; history of blood transfusions during antineoplastic treatment; no previous history of multiple transfusions preceding diagnosis of leukemia; ferritin concentration obtained in numerous time points during treatment; no clinical signs and no laboratory markers of infection and inflammatory process during SF evaluation. Exclusion criteria were as follows: patients with acute leukemia who underwent only one blood transfusion throughout treatment; all patients with oncological diagnoses other than leukemia; inflammatory process and acute infection in the moment of SF evaluation. The data from patients’ transfusions books, physical examinations, and medical documentation were analyzed, from January 2010 until December 2019. The following variables were collected from all patients: demographics (age and gender), clinical diagnosis (diagnosis, date of diagnosis). After excluding non-leukemic patients results from remaining 135 participants (male: 77, female: 58) between 0.06 and 17.6 years of age (median age 5.62 years) were analyzed. We performed precise analyses in subgroups according to diagnosis—ALL (*n* = 110) and AML (*n* = 25). The population was also divided into 4 subgroups according to age (Table 1 and Table 2a,b). The study received approval from the Local Ethical Committee of Medical University of Bialystok, permission number: APK.002.36.2021.

### 2.2. Laboratory Tests

Serum ferritin (SF, ng/mL) was used as a marker of iron overload (IO). SF was measured using the electrochemiluminescence method in every patient. In all oncological patients, measurements were carried out at different stages of therapy: at the beginning of the treatment (single measurement—SF at baseline), in the course of the treatment (5 separated measurements during the treatment depending on the time of the therapy TP1-TP5, mean interspace between each subsequent point was 2–4 months) and after cessation of therapy (single measurement at the end of treatment—SF at finish). The cutoff categories describing IO according to SF elevated (>500 ng/mL) and significantly elevated (>1000 ng/mL) are widely used in the literature regarding the risk of post-transfusion IO [25,26,27,28]. All of the measurements were performed in the local laboratory in Children University Hospital in Bialystok.

All the baseline characteristics were collected from the medical records, including age, sex, diagnosis, and serum ferritin concentration (ng/mL). The diagnosis of neoplastic disease was based upon evaluation of peripheral blood blast percentage, bone marrow blast percentage, and immunophenotype. The laboratory data were collected within the first 24 h after the admission.

### 2.3. Blood Transfusion

The level of hemoglobin at which PRBCs (packed red blood cells) were administered was below 8 g/dl taking into account the present clinical symptoms of anemia and age of the child. In most patients, transfusion was performed in cases of lower than 8 g/dl. In terms of blood transfusions, the parameters were measured in each patient as follows: (A) total amount of received blood in milliliters (during whole treatment process), (B) total amount of received blood per kilogram of body weight, and (C) total number of received transfusion units—episodes of transfusion.

### 2.4. Statistical Analysis

Processing of the acquired data was performed with the use of GraphPad Prism 9.0 software (GraphPad Prism Inc., San Diego, CA, USA). The normality of the data distribution was verified before each analysis, implemented using Shapiro–Wilk, Anderson–Darling and D’Agostiono–Pearson tests. Differences between the unpaired groups were analyzed using a nonparametric Mann–Whitney test. Chi-square tests were applied to assess the differences in the selected parameters distribution among the tested groups. The significance of the changes reported in the course of therapy monitoring was established with two-way ANOVA, with Fisher’s LSD test. The association between ferritin and other clinical and laboratory parameters was analyzed with the use of nonparametric Spearman correlation test. For survival analysis, the log-rank (Mantel–Cox) test was applied. The significance level was set at *p* < 0.05, indicated within the manuscript with exact *p* value or asterisks: *—*p* < 0.05, **—*p* < 0.01, ***—*p* < 0.001, ****—*p* < 0.0001. 

## 3. Results

### 3.1. Blood Transfusions

A total of 100% of patients received blood transfusions, with the lowest number of two episodes of PRBC. The data presented in Table 3 give a detailed description of the transfusion parameters within the whole group and subgroups. Patients treated for leukemia received, on average, 128.6 mL of blood per kilogram of body weight (ranging from 16.7 mL/kg to 1800 mL/kg; mean 171.29) and 17.62 transfusions (2–78 units; median of 14 units). A total of 98 patients (72.59%) received at least 100 mL of PRBC per kg, including 77 patients with ALL (70%) and 22 with AML (88%). The maximum value showed in the table refers to a 15-year-old female patient who was treated primarily for acute lymphoblastic leukemia and then for acute myeloblastic leukemia as a second neoplasm; the patient received a total of 78 transfusions throughout the treatment, totaling 23,400 mL of blood—1800 mL per kilogram of body weight. With the assumed significance level, for all three variables (A–C), no statistically significant difference was found between the median values between the groups according to gender overall and gender groups between AML and ALL (*p* > 0.05) (Table 3).

### 3.2. Ferritin Levels in AML and ALL Patients before Therapy

No significant differences were observed between AML and ALL patients in the context of plasma ferritin levels before therapy implementation or when additional stratification was performed into no graft subjects and group with bone marrow allograft (Figure 1A). Regarding the sex of the analyzed patients, only in the ALL group were reduced ferritin concentrations reported in male versus female subjects (*p* = 0.0424). In addition, the same leukemia groups demonstrated age-based differences in the tested parameter. Patients with ALL within range of 5 to 10 years showed lower ferritin levels compared to children younger than 5 years (*p* = 0.0058) and older than 10 years (*p* = 0.0012) (Figure 1B).

Next, we assessed the distribution of the subjects with lower and higher plasma ferritin values (based on the total median) within the studied groups. In accordance, we did not show any differences between AML and ALL patients. Additional sex-based grouping did not introduce changes to that status in either the AML or the ALL subjects. Significant differences in lower/higher ferritin levels were further shown in the context of the various age ranges. Patients with AML who were older than 10 years demonstrated a tendency for increased ferritin levels compared to children aged below 5 years (*p* = 0.0734). A similar situation was observed in the ALL patients, where subjects older than 10 years showed a higher incidence of plasma ferritin versus leukemic patients younger than 5 years (*p* = 0.0144) (Figure 1C).

### 3.3. Blood Transfusion Effects on Ferritin Levels in AML and ALL Patients

Considering the assumed cutoff levels for SF at the start of the treatment, percentage of children with mildly elevated SF (500–1000 ng/mL) and significantly elevated SF (>500 ng/mL) were calculated. In the whole group, 40.74% (*n* = 55) of patients had increased SF > 500 ng/mL; the results were similar in both the ALL and the AML subgroups (40 vs. 44%). A proportion of 14% (*n* = 19) of the whole group was diagnosed with serum ferritin higher than 1000 ng/mL; this accounted for 12% of AML patients and 14.54% of ALL patients. The analysis of the patients who finished treatment with elevated SF showed that 80% of them finished therapy with an SF higher than 500 ng/mL; in the AML group this percentage was 96%. With the cutoff point at 1000 ng/mL at the cessation of therapy, we found that 31.11% of patients had this (28,18% with ALL and 44% of AML). Comparison of patients with increased SF both at the beginning and at the cessation of treatment showed this tendency in both groups regardless of the diagnosis. Our analysis showed that the majority of patients who were diagnosed with high SF finished treatment with high SF (Table 4).

The analysis of all the AML and ALL patients showed a significant increase in plasma ferritin levels from the very beginning of therapy. At T2–T3 points of the monitoring phase, those levels seemed to be stabilized without any further increase. It is noteworthy that the last time points demonstrated significantly higher ferritin levels in the AML patients (*p* = 0.0319). Such a tendency was also observed earlier at T1 (*p* = 0.0545) (Figure 2A). Similar changes in time were observed in the AML group when assessing the differences between the female and male subjects. However, unlike males who maintained significantly higher levels of plasma ferritin, concentrations in female AML patients seemed to be maintained on the verge of statistical significance compared to the first time point. In the ALL group, a constant increase in ferritin level was reported until T3, where male subjects showed significantly higher levels compared to females (*p* = 0.0160). Despite further decline in ferritin concentrations, the values were still significantly higher than those at T0, with differences between male and female subjects maintained until T5 (*p* = 0.0227) (Figure 2B). Considering age-based stratification, AML patients aged below 5 and above 10 years demonstrated a stable increase in plasma ferritin until the T3–T4 time points. The achieved levels were maintained at similar levels through the rest of the monitoring time points. Interestingly, subjects with AML aged between 5 and 10 years did not reach significantly higher levels of ferritin throughout the period, with only a slight peak showing at the early T2 stage (*p* = 0.0738). In reference to the ALL patients, all age groups managed to achieve a significant increase in ferritin concentration, with highest peak demonstrated at the T3–T4 time points. Despite slight further decline, all the subjects maintained higher levels of the monitored parameter. It is noteworthy that the most essential changes were reported in ALL patients older than 10 years. Those subjects, at several time points (T2, T3, T5, and T6), showed values that were substantially higher than those of other groups (at T5: *p* < 0.0001) (Figure 2C).

As a next step, we verified the eventual associations between the ferritin levels in the time and clinical or laboratory parameters. First, we demonstrated a positive moderate correlation of ferritin change with the age (AML: *p* = 0.012; ALL: *p* = 0.002) and the weight (AML: *p* = 0.021; ALL: *p* = 0.002) of the subjects. No association was found in reference to the transfused blood volumes; however, the ferritin levels in the course of therapy were found to be positively correlated with the units of blood per kilogram: delta T6 versus T0 in AML patients (*p* = 0.0039) and subjects with ALL at T6 (*p* = 0.0003). Interestingly, further analysis revealed that plasma ferritin at T0 correlated positively with CRP (*p* = 0.005) and PCT (*p* = 0.009), but that association seemed to be lost in the course of therapy. In contrast, AML subjects demonstrated a negative correlation of T0 ferritin with PCT; however, no significance level was achieved. The values of the ferritin reported further in the course of therapy showed a strong positive association with PCT, and a moderate negative association with CRP (no significance level achieved) (Figure 2D).

### 3.4. Blood Transfusion Volumes and Units among AML and ALL Patients

The patients with AML received significantly higher blood volume compared to the ALL subjects (*p* = 0.0003). That difference was maintained when only the patients without graft (*p* = 0.0007) or with bone marrow allograft (*p* = 0.0322) were analyzed (Figure 3A). No crucial changes were reported when sex- or age-based stratification of the patients was applied. A slight exception was a tendency for lower volume of blood transfused in the AML subjects older than 10 years compared to those within the range of 5 to 10 years (*p* = 0.0824). Similarly, ALL patients of the same age had significantly lower volumes of blood used versus children below 5 years (*p* = 0.0151) (Figure 3B).

Considering transfused blood units, the AML subjects demonstrated higher usage (*p* < 0.0001), with the same difference when assessing no graft groups (*p* < 0.0001). No difference, however, was demonstrated between AML and ALL patients who underwent bone marrow transplants (*p* = 0.1111) (Figure 3C). Contrary to what was noted in the analysis of the transfused blood volume, female subjects had more units per kilogram used compared to males in the AML group (*p* = 0.0020). The same leukemic group showed higher blood units implemented among patients older than 10 years versus those between 5 and 10 years (*p* = 0.0420) and younger than 5 years (*p* = 0.0004). No changes were found in reference to the ALL patients in terms of both sex and age (Figure 3D).

### 3.5. Ferritin Levels and Blood-Transfusion-Related Parameters Influence on the Outcome of the AML and ALL Patients’ Therapy

The relative risk analysis revealed that AML patients with a low initial level of serum ferritin demonstrated 2.76 times higher risk of death compared to subjects with a high level (*p* = 0.0446). Additional stratification based on sex and age revealed that such a tendency is reported in male patients (RR = 5.00; *p* = 0.0308), predominantly those younger than 5 years old (RR = 4.50; *p* = 0.0651). No significant changes were observed in the ALL subjects. However, females (RR = 0.38; *p* = 0.1748) and children below 5 years old (RR = 0.31; *p* = 19.26) with lower ferritin levels seemed to have a reduced risk of death (Figure 4A). Unlike ferritin levels, the volume of the blood for transfusion did not seem to have a crucial effect on the death occurrence in AML patients. Only in AML subjects older than 10 years old with lower volumes of the transfused blood was a tendency for higher risk of death found (RR = 3.75, *p* = 0.0989). Interestingly, patients with ALL were found to demonstrate significantly lower death risk among subjects aged between 5 and 10 years old and where lower volumes of blood were used for transfusion (RR = 0.09; *p* = 0.0132) (Figure 4B). Similar results were obtained when we analyzed the influence of the transfused blood units. Despite no significancy, AML patients older than 10 years old also showed higher risk of death incidence (RR = 2.67; *p* = 0.2357). As described above, ALL patients aged 5 to 10 years old also showed a reduced risk of death in the group with lower units of blood used in therapy (RR = 0.10; *p* =0.0188) (Figure 4C). In addition, similar effect of lower units of blood used was observed in female ALL subjects (RR = 0.25; *p* = 0.0416) and all the ALL patients (RR = 0.41; *p* = 0.0703) (Figure 4C).

The subsequent survival assessment demonstrated essential differences in survival curves between patients with low/high pre-treatment serum ferritin levels. In the AML group, subjects with higher ferritin levels showed longer survival (median = 1132 days) compared to the opposing group (median = 117 days) (HR = 2.67 [low versus high]; *p* = 0.0854). In the case of the ALL patients, lower initial ferritin concentrations were associated with better survival (median = 1405 days), with significantly reduced values in the other group (median = 236) (HR = 3.57 [high versus low]; *p* = 0.0027) (Figure 4D). We did not observe significant differences in survival between patients treated with low/high volume of transfused blood, especially in the group of patients with ALL. Subjects with AML and higher volumes of the blood transfused seemed to have higher chances of survival (HR = 2.62 [low versus high); *p* = 0.1289) (Figure 4E).

## 4. Discussion

The estimation of iron overload using evaluation of serum ferritin does not seem to be fully objective; however, this remains the easiest and most accessible method. Since ferritin is an acute phase reactant, it may be falsely elevated due to any ongoing systemic inflammatory processes and thus may not be the most reliable measurement of body iron burden [29]. Studies evaluating the role of ferritin in cancer showed a statistically significant increase in SF in both localized and metastatic diseases [7,30,31,32]. Serial measurements of ferritin concentration performed during anticancer therapy with chemotherapy showed normalization of SF levels with the proper therapy response [33,34,35]. On the other hand, increase in SF is often a sign of progression and poor survival [30]. Most pediatric leukemia patients receive combination chemotherapy over many months or years, and it is extremely rare for a patient to complete their therapy without PRBC transfusions. Information regarding the risk of cumulative erythrocyte transfusions and tissue iron accumulation is well understood among non-malignant populations. However, very little is known about transfusion-related iron burden among oncology patients, especially in the pediatric population [29]. Secondary anemias requiring transfusion can occur not only as a result of the primary disease and its treatment, but also due to unintentional blood loss during frequent laboratory analyses, especially at the beginning of therapy. The volume of blood taken for analysis can be especially significant among infants and younger children, in certain cases even leading to the need for transfusion. In order to minimize blood loss, strategies should be introduced and widely popularized for chronically ill hospitalized patients, including in hematology and oncology units [36].

In many analyses, the cutoff level of ferritin defining IO was described as 1000 ng/mL or even 1500 ng/mL [4,8,27,37]. Our analysis was based on the serum ferritin level in the pediatric population; so, in accordance with the recommendations of Trovillion et al. and Wood et al., we decided to analyze two cutoff points—the first at 500 ng/mL and the second at 1000 ng/mL. We used 1000 ng/mL due to it being traditionally used in the definition of IO and as a cutoff level for qualification for chelation therapy [8,27,28]. However, the results of the study by Kim et al. demonstrated that even lower concentrations of ferritin are efficient for predicting patients’ outcomes. Kim et al. showed that high and high–intermediate risk of international prognostic index and ferritin level (≥500 ng/mL) were independent poor prognostic factors for PFS and OS among patients with lymphoma. Among patients with lower IPI scores, a high ferritin level was found to be an independent poor prognostic factor for PFS and OS [38]. In our analysis, we proved that both 500 ng/mL and 1000 ng/mL, at the onset of neoplastic treatment, were connected with diagnosis and not with disturbances in iron metabolism. Interestingly, our data showed that AML patients with a lower ferritin level had significantly higher risk of death compared to high-level subjects. This finding seemed to be predominantly associated with male subjects and children younger than 5 years. The analysis of the survival curves revealed that AML patients with higher ferritin levels at diagnosis have higher chances of survival. In contrast, patients with high ferritin levels while being diagnosed with ALL exhibited worse prognoses.

Lower volumes of blood transfused were only associated with lower death risk in ALL subjects compared to the high-level subgroup. Similar changes were reported in reference to blood units implemented, with an additional reduced risk of death incidence reported in females with lower transfused units. No statistically significant differences were demonstrated in the context of transfused blood units’ levels and survival. However, AML patients with more units transfused seemed to survive longer than their counterparts with less blood transfused; however, this result was not found to achieve statistical significance. Our results oppose those obtained by Ihlow et al., who showed that high SF had a negative impact on long-term survival in 137 adult patients with AML. We explain this discrepancy through reference to the small number of subjects and possible differences in outcomes between children and adults [39].

We found that more than 40% (*n* = 55) of all patients had increased SF > 500 ng/mL, and 14% (*n* = 19) had over 1000 ng/mL at the beginning of treatment. At the cessation of the treatment, 80% (*n* = 110) of children presented with SF over 500 ng/mL and more than 31% (*n* = 42) presented with SF higher than 1000 ng/mL. Comparison of patients with increased SF at both points—the start and the end of therapy (96.35% with SF > 500 ng/mL and 73.68% with SF > 1000 ng/mL)—proved that starting treatment with high SF is connected with high SF at the end of the treatment.

High serum ferritin levels, without any correspondence to the amount of total body storage iron, have been found among patients with leukemia. Investigating 96 adults with different types of leukemia, Aubert et al. found that serum ferritin can be used as a tumor marker in myeloid leukemias. Extremely high serum ferritin levels were seen in acute myeloblastic leukemia before treatment and in blastic crisis of chronic myeloid leukemia (i.e., 21-fold increased serum ferritin concentrations). Among patients with acute myeloblastic leukemia who achieved complete remission, ferritin decreased to a normal level [40]. Our analysis did not show a significant drop in SF during treatment among AML patients, but this may be attributable to the small number of patients.

We also analyzed a group of 14 patients who started oncological therapy with a ferritin concentration higher than 1000 ng/mL and received more than 100 mL/kg body mass PRBC; this comparison was performed to determine whether these patients were more prone to developing iron overload. We also sought to answer the following question: can they be considered a marker of leukemic growth or of post-transfusion iron overload? The analysis showed that high SF at the start of therapy among leukemia patients who received more than 100 mL/kg of PRBC is not a predictor of high SF at the end of therapy. On the other hand, receiving more than 100 mL of PRBC does not aggravate primarily increased SF ferritin. Our analysis of this specific group of patients also proved that knowledge about the volume of transfused PRBC can be considered more of an objective marker of IO than SF itself. In their analysis of 139 patients with leukemia, Cacciotti et al. found that 23% of all the patients had post-treatment ferritin concentrations of over 1000 ng/mL; those with HR-ALL and AML were more likely to be transfused with >=10 RBC units and were more prone to develop hepatic dysfunction. In our analysis, the lowest number of episodes of transfused blood was 2, while the highest was 78. The number of transfusions was the highest among the patients with AML, HR-ALL, and HSCT; the number of transfusions was correlated with SF after treatment and delta among the patients with both ALL and AML [41,42]. An additional analysis revealed reduced blood volumes used for transfusion among ALL patients compared to AML patients. That difference was also maintained when separately analyzing the subjects with no graft or with bone marrow allograft. Additional sex- and age-based stratification revealed slightly lower blood volumes transfused in ALL patients older than 10 years. Despite similar results in the context of transfused blood units per kilogram, we did not find differences between AML and ALL patients who underwent bone marrow allografts. Here, moreover, male AML subjects demonstrated reduced transfused blood units, whereas the same leukemic patients older than 10 years old showed higher numbers of transfused blood units.

According to Cacciotti et al., the number of transfused units can be a better predictor of cardiac and hepatic iron overload than serum ferritin evaluation. According to some data qualification for MRI T2 screening of liver and heart, even only upon either a history of erythrocyte transfusion ≥10 times or serum ferritin ≥1000 ng/mL or both, the number of transfusions itself is better predictor of IO than SF. It is noteworthy that patients with SF lower than 1000 ng/mL and with more than 10 RBC transfusions were found to have 27.3% of hepatic and 18.2% of cardiac iron loading [41].

The time of the analysis of SF seems to be the main factor explaining the phenomenon of the ferritin increase. The cancer-associated elevation in serum ferritin at diagnosis is most likely induced by an inflammatory state and not due liver damage or other alterations in the body’s iron stores [1]. The exclusion of the influence of coexisting inflammatory processes on the incidence of hyperferritinemia in the moment of diagnosis allowed us to take into account the oncological explanation of high ferritin concentration. It is noteworthy that the measurements of SF during and at the end of therapy were performed in the absence of concomitant inflammation. Many analyses in the literature are based on SF measurement after the cessation of treatment; meanwhile, our study shows dynamic changes in ferritin concentration through anticancer therapy. We found a strong positive correlation between SF at the baseline and finish point, which means that patients with high concentrations of ferritin at the moment of diagnosis are prone to finishing treatment with high levels of ferritin. The reason for this increase seems to vary, with nonspecific markers of neoplastic disease at the beginning and markers of IO at the end of treatment. Considering our data, in both the AML and ALL patients, time-dependent changes in ferritin were found to positively correlate with the age and weight of the studied children. Those variations in ferritin also correlated with the transfused blood units per kilogram, predominantly in the ALL group. The initial ferritin levels (at T0) correlated positively with CRP and PCT; however, that association was diminished during the course of therapy. In contrast, ferritin at diagnosis in the AML group seemed to correlate negatively with PCT, but changed towards strong positive association over time. Changes in ferritin were also associated with their negative correlation with CRP in AML; it is noteworthy that no statistical significance was achieved.

Specific monitoring of ferritin concentration during oncological therapy could allow for differentiation between the incidence of initial cancer-caused hyperferritinemia and that related to blood transfusion throughout treatment. Induction therapy with a satisfactory response to treatment should lead to transient normalization of SF and subsequent increase due to transfusion-related iron overload, but the recognition of this specific time point can be difficult and hard to capture. In our analysis, we did not observe the moment of normalization of SF during therapy, which could be caused by the overlapping early results of RBC transfusions, especially in cases with initially low, life-threatening hemoglobin levels.

The disturbed iron metabolism and strongly increased ferritin concentrations not only in the serum but also in the cancer tissue seem to be responsible for the elevated levels of ferritin in the cancers. Increased serum ferritin levels at the cessation of treatment are connected rather with multiple transfusions, especially among patients treated for leukemias and MDS, with the highest risk among those who have undergone stem cell transplantation. The link between the first and the last analyzed ferritin concentration remains unclear. On the other hand, some of the patients are prone to having higher SF at diagnosis, during the treatment, and after the cessation of the treatment. They develop iron overload more easily than others, even with comparable transfused blood volumes and the same type of oncological disease. The genetic background of this phenomenon, including HFE gene mutations, should be also taken into consideration during qualification for multiple transfusions to avoid the additional increase in iron burden [43]. Increasing serum ferritin level during long-time leukemia therapy and its regular measurement should help in distinguishing a group of patients for the use of chelators alongside oncological therapy or directly after cessation. Nowadays, oral chelators such as deferasirox are used with a satisfactory effect among children with iron overload due to leukemia, MDS, thalassemia, or Blackfan–Diamond anemia [44,45]. However, conducting chelation therapy alongside oncological treatment can be questionable due to the concomitant side effects described in the literature, primarily concerning the kidneys and the liver; such an approach is not currently recommended in to treatment protocols.

Amid et al. showed that children diagnosed at an early age, before 10 years of age, with lower body surface area (BSA) and the potential to achieve an increase in BSA, had lower risk of transfusion iron overload during treatment than those who were diagnosed as teenagers and had already completed their growth spurt [46]. Unal et al. confirmed that the time of ALL diagnosis may affect long-term iron loading among these patients. According to the data, growth spurts may cause a decline in the stored iron among patients who were diagnosed with ALL before 10 years of age, which can be related to increased demand for iron due to the expansion in blood volume [8]. Our analysis also showed that the ferritin concentration, the number of transfusions, and the volume of blood transfused (measured in ml/kg) was the highest among children older than 10 years. In the youngest children—below 12 months—ferritin concentration remained higher than those from groups 2 and 3. The explanation of this result lies in there being a small number of patients who were younger than 12 months and who had a physiologically higher level of serum ferritin, as is the case for newborns and younger neonates—4/8 patients were younger than 3 months.

The therapy for AML is aggressive and therefore leads to severe side effects, including the need for more frequent transfusions. According to Nottage et al., recipients of conventional therapy due to AML had a significantly higher transfusion burdens compared to those with conventional ALL treatment—i.e., a median of 21 vs. 6 transfusions, 3793 mL vs. 1134 mL, and 182 mL/kg vs. 57 mL/kg (all *p*-values < 0.001), respectively [29]. In our analysis, patients with AML had statistically higher serum ferritin at both the start and the end points of the treatment compared to patients with ALL; they received more blood units and had more episodes of PRBC transfusions. Comparing the ALL patients from the HR and the non-HR groups, we found that those treated with more intensive protocols received more PRBC (in ml) per body weight (in kg) and had more transfusion episodes. Children from the HR-group finished with statistically higher serum ferritin concentrations than those from the non-HR group. Additional investigation showed that ferritin increased significantly in the course of the therapy, reaching peak values usually around T3–T4. A similar direction of changes was observed in both the studied groups; however, at certain time points (T1 or T5), patients with AML showed significantly higher concentrations than patients with ALL. From T4, males seemed to maintain higher levels of ferritin compared to female subjects in the AML group. That difference was statistically significant in reference to the ALL group, divided into males and females, starting from the peak values at T3. AML subjects aged 5–10 years, unlike other subgroups, did not achieve a statistically significant increase in ferritin levels. All age-based subgroups of the ALL group achieved a constant increase in ferritin until T3, with a subsequent slight decline in that parameter. It is noteworthy that, during most of the monitoring time, the ALL subjects who were older than 10 years dominated over other subgroups in terms of ferritin concentration.

The analysis performed by Lecka et al. on 71 patients with ALL and AML showed that a higher concentration of ferritin was connected with worse overall and event-free survival and a higher incidence of relapse. Lecka et al. showed that a ferritin concentration higher than 1000 ng/mL is a strong adverse independent marker of survival for children treated for acute leukemia, both with and without hematopoietic cell transplantation (HCT). Lecka’s analysis corroborates the results presented here. Similarly, we proved that serum ferritin level at the end of therapy (finish SF) is a maker of worse prognosis in leukemic pediatric patients; moreover, high delta SF (the difference between SF at the start and end of treatment) proved to be a strong negative prognostic marker for relapse and death [47].

We are aware of the limitations in our study. An important limitation of our analyses is in our defining iron burden in terms of serum ferritin levels. Analysis of ferritin is inexpensive, and it is a widely available indicator of IO; however, a more accurate measure in assessing iron content in the body is the LIC (liver iron concentration) measurement that is performed through MRI imaging. This is more sensitive and specific and less invasive; however, it is more expensive and less accessible than assessing ferritin. However, the correlation between plasma ferritin and LIC in pediatric patients has been investigated and both of these measurement methods have proven to be highly correlated [18]. However, the analysis identified a group of patients for whom further LIC screening should be considered. Another major disadvantage seems to be the measurement of ferritin without the simultaneous analysis of CRP values; the increase of these could suggest the presence of inflammation and thus also explain the increased concentration of SF. We have not noticed other coexisting medical conditions that could disturb the concentration of ferritin, such as haemochromatosis or ascorbate deficiency. Further disadvantages lie in the small research group and the small number of diagnosis groups and age groups.

## 5. Conclusions

In summary, iron overload, as measured through serum ferritin concentration, is a common side effect of antineoplastic treatment among children and young adults. We showed that different ages among the AML and ALL groups can affect the ferritin levels for both the basal level and its response to the applied therapy. In fact, changes in serum ferritin levels were significantly related to the patients’ age and weight, together with the volume of the blood used for transfusion. SF at the start of the treatment seems to be a good, nonspecific tool for diagnosis of leukemia; however, its role in monitoring the response to oncological treatment can be disrupted by the overlapping effect of blood transfusions. Importantly, we demonstrated that lower pre-treatment values of the ferritin are associated with better survival among ALL subjects. A different tendency was found in patients with AML who had high basal ferritin concentrations, leading to a lower death rate. Further investigation is of great importance in establishing the mechanism behind the various effects of initial serum ferritin levels on subsequent therapy-related survival among studied patients.

## Figures and Tables

**Figure 1 cancers-16-00367-f001:**
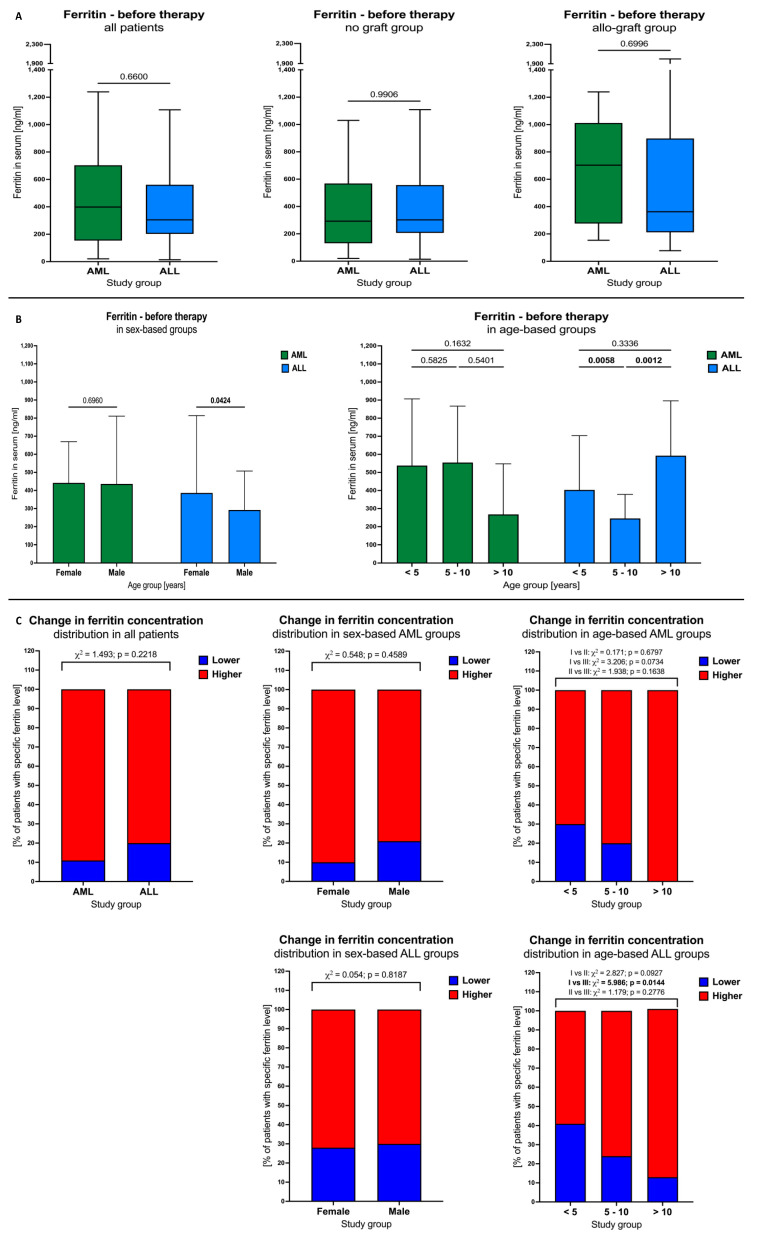
Differences in blood ferritin levels in the studied AML and ALL patients. Initial ferritin concentrations between the total group of AML and ALL subjects, those with no graft and patients with allo-graft transplant (**A**). Influence of sex and age (years) on basal ferritin levels within leukemic groups (**B**). Lower and higher ferritin concentrations (based on the median value) within the tested subgroups: total patients, sex- and age-based (years) (**C**). Data presented as median values and 25th–75th percentile (**A**,**B**); percentage share of low/high ferritin levels in selected groups (**C**).

**Figure 2 cancers-16-00367-f002:**
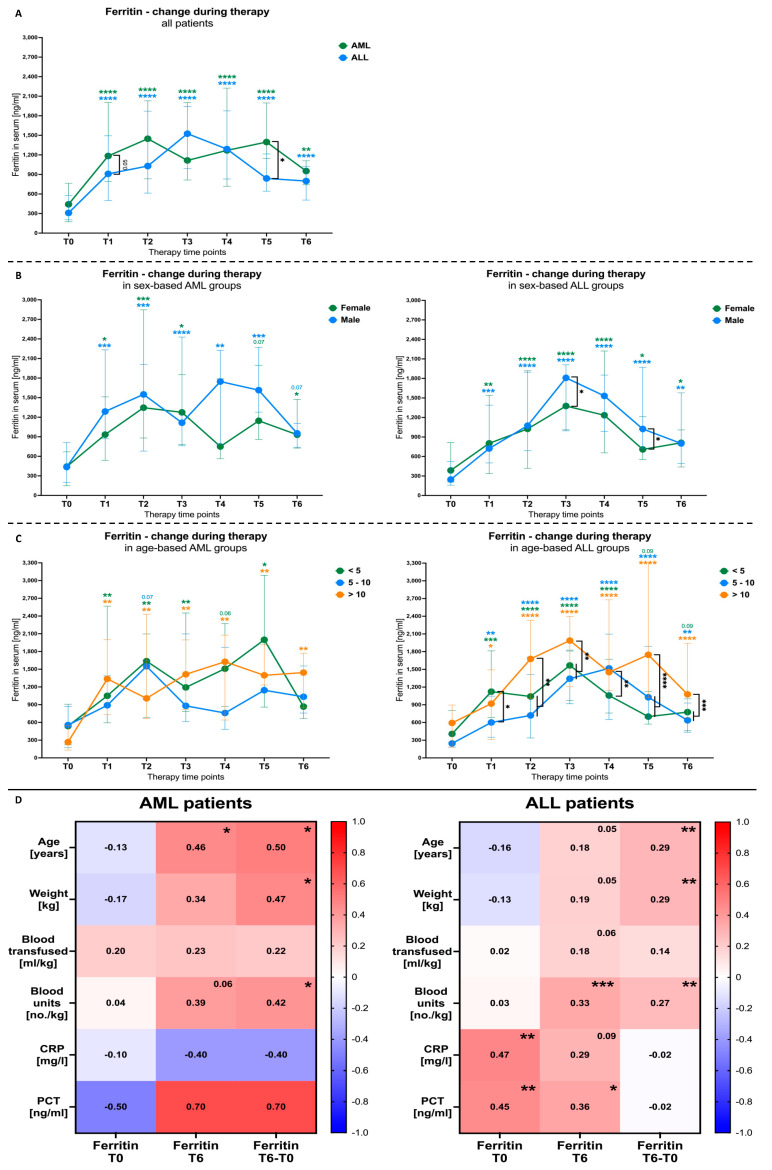
Blood-transfusion-induced changes in ferritin level of AML and ALL patients. Changes in serum ferritin levels in total ALL or AML group (**A**), including stratification based on patients’ sex (**B**) and age (years) (**C**). Correlation of ferritin, at selected periods of the therapy monitoring, with clinical and laboratory data (**D**). Data presented as median values and 25th–75th percentile (**A**–**C**); and heat maps with correlation coefficient (r) values (**C**). Colored asterisks correspond directly to the graphed groups and indicate differences between selected time points versus beginning of the therapy (T0). The levels of significant differences were indicated with asterisks or exact *p* values: * *p* < 0.05; ** *p* < 0.01; *** *p* < 0.001; **** *p* < 0.0001.

**Figure 3 cancers-16-00367-f003:**
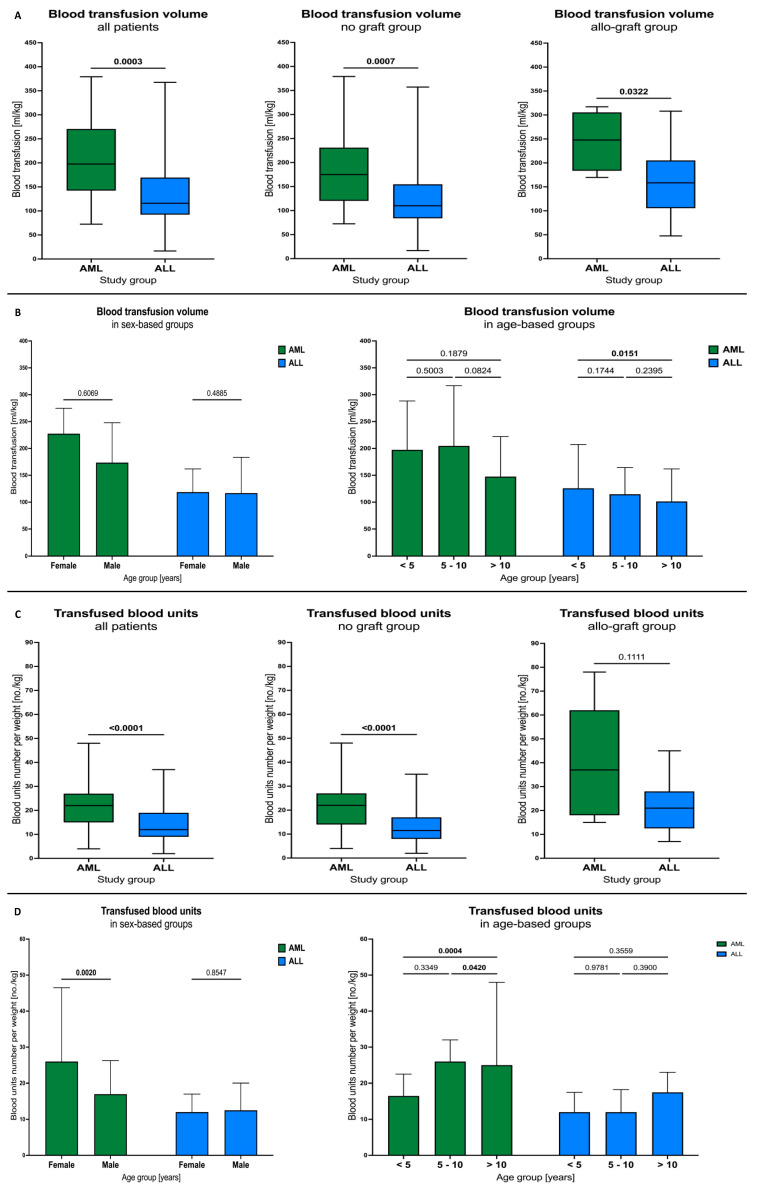
Differences in transfused blood volumes and units between the studied AML and ALL patients. Blood transfusion volumes in AML and ALL subjects, those with no graft, and patients with allo-graft transplant (**A**). Influence of sex and age (years) on blood transfusion volumes within leukemic groups (**B**). Blood transfusion units in AML and ALL subjects, those with no graft, and patients with allo-graft transplant (**C**). Influence of sex and age (years) on blood transfusion units within leukemic groups (**D**). Data presented as median values and 25th–75th percentile.

**Figure 4 cancers-16-00367-f004:**
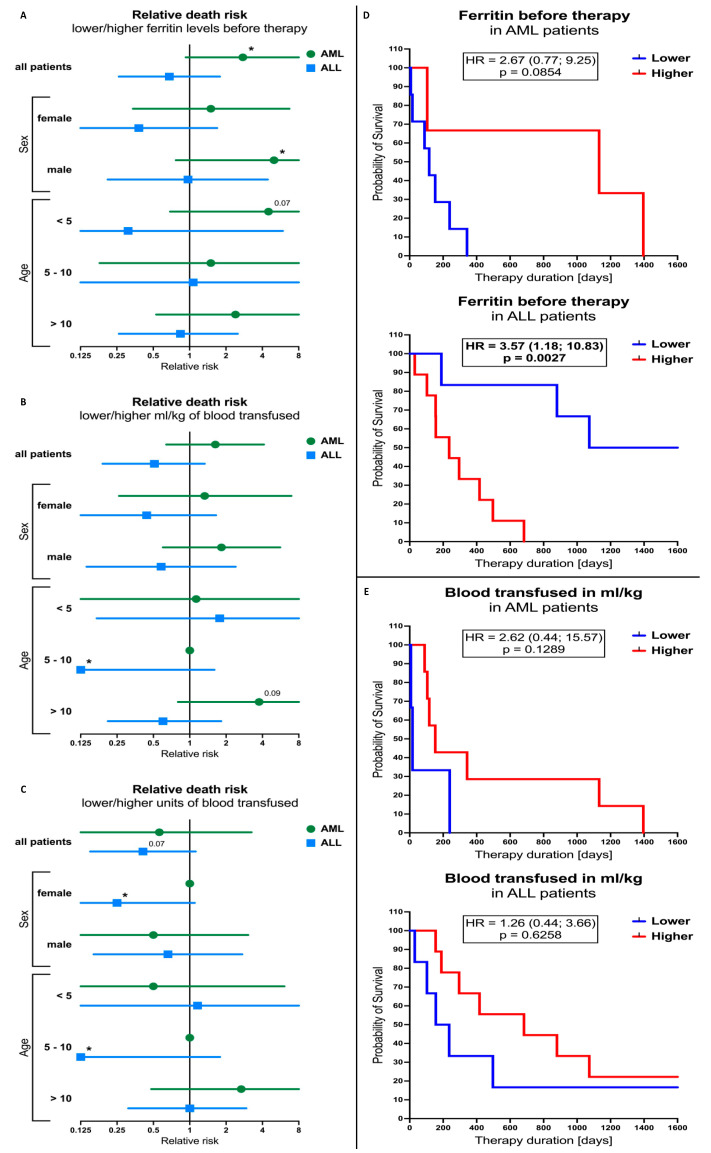
Significance of ferritin levels and blood-transfusion-related parameters on AML and ALL patients’ survival. Analysis of relative risk of death incidence between low/high serum ferritin (**A**), low/high transfused blood volume (**B**), or low/high transfused blood units (**C**) groups, in AML and ALL subject, including sex- and age-based (years) stratification. Survival curves of AML and ALL patients in context of ferritin levels (**D**) and volumes of blood transfused (**E**). Data presented as relative risk values with confidence interval (**A**–**C**); survival percentage during therapy, with log-rank test results (**D**,**E**). The levels of significant differences were indicated with asterisks or exact *p* values: * *p* < 0.05.

**Table 1 cancers-16-00367-t001:** Characteristics of the patient groups in terms of age and gender.

	LEUKEMIAS			ALL *			AML **		
	Total	Males (M)	Females (F)	Total	M	F	Total	M	F
*n*	135	77	58	110	64	46	35	23	12
%	100	57.03	42.97	100	58.2	41.82	100	65.71	34.28
Median (years)	5.62 yrs	6078	8127						
	0.06–17.6								

* ALL Acute lymphoblastic leukemia; ** AML Acute myeloblastic leukemia.

**Table 2 cancers-16-00367-t002:** (a) Division into age subgroups; (b) Study group structure in terms of diagnosis and age.

(a)						
	AGE GROUPS					
	≤1 yr Group 1			1–5 yrs Group 2	5–10 yrs Group 3	≥10 yrs Group 4
Total	8 8/135 30.06%			51 51/135 37.78%	43 43/135 31.85%	33 33/135 24.44%
ALL	3 3/110 2.72%			46 46/110 41.81%	38 38/110	23 23/110 21.82%
AML	5 5/25 20%			5 5/25 20%	5 5/25 20%	10 10/25 40%
**(b)**						
	**Protocol**	** *n* **	**Allo HSCT**	**REC**	**Deceased**	**Second** **Malignancy**
ALL	ALLIC 2009	51	25	4	9	1
	ALLIC 2012	59				
AML	AML BFM 2002	25	23	4	10	0

**Table 3 cancers-16-00367-t003:** Transfusion-related parameters according to the diagnosis.

	Median	Mean *	SD *	Min	Max
**LEUKEMIAS (*n* = 135)**					
A. Total amount of transfused blood [mL]	3300	4311.67	3443.45	360	23,400
B. Total amount of transfused blood per kilogram of body weight [mL/kg]	128.6	171.29	173.65	16.7	1800
C. Total number of transfusions [units]	14	17.62	11.99	2	78
**ALL (*n* = 110)**					
A. Total amount of transfused blood [mL]	3300	3869	2624	360	11,100
B. Total amount of transfused blood per kilogram of body weight [mL/kg]	121.4	150.4	103.1	16.70	532.9
C. Total number of transfusions [units]	12	15.79	10.13	2	49
**AML (*n* = 35)**					
A. Total amount of transfused blood [mL]	5035	6359	5552	725.0	23,400
B. Total amount of transfused blood per kilogram of body weight [mL/kg]	201.2	267.8	336.2	72.50	1800
C. Total number of transfusions [units]	22.50	26.08	16.01	4	78

* The was no normal distribution in the assessed data, mean and SD were posted for informational purposes.

**Table 4 cancers-16-00367-t004:** Characteristic of patients with SF > 500 mg/mL and SF > 1000 ng/mL at the beginning at the of the treatment.

Serum Ferritin Start >500 ng/mL *n*	Median	Serum Ferritin Finish >500 ng/mL *n*	Median	Same Patients *n*
Total 55 55/135 (40.74%) ALL 44/110 (40%) AML 11/25 (44%)	889 712–963	Total 108 110/135 (80%) ALL 86/110 (78.18%) AML 24/25 (96%)	933.7 500.1–4945	53 (96.35%)
**Serum Ferritin Start** **>1000 ng/mL** ** *n* **	**Median** **Min–Max**	**Serum Ferritin Finish** **>1000 ng/mL** ** *n* **	**Median** **Min–max**	**Same Patients** ** *n* **
Total 19 19/135 (14.07%) ALL 16/110 (14.54%) AML 3/25 (12%)	1692 1097–2127	Total 42 42/135 (31.11%) ALL 31/110 (28.18%) AML 11/25 (44%)	1589 1001–4945	14 (73.68%)

## Data Availability

The data presented in this study are available in this article.

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
