# Peer review of "Iron Overload in Children with Acute Lymphoblastic and Acute Myeloblastic Leukemia—Experience of One Center"

_cancers, 2024, doi:10.3390/cancers16020367_

Round 1
Reviewer 1 Report
Comments and Suggestions for Authors
In the manuscript entitled “Iron overload in children with acute leukemia – experience of one center”, by Malgorzata Sawicka-Zukowska and colleagues, authors investigate the effect of serum ferritin levels, as an indicator or iron overload, on outcome of the patients with AML and ALL. The main conclusions are that the high ferritin levels before treatment in patients with AML and low levels in patients with ALL confer better prognosis. Overall manuscript is very well written and informative. However, several methodological issues hamper this study, among which the main issue remains the sole usage of ferritin as a marker of iron overload. As authors themselves acknowledge, this is not a reliable marker of iron overload and thus statements suggesting this throughout the manuscript and in the title of the article are misleading. Other major issues:
1) Authors do not show data about infections and inflammations during the treatment. Since these are quite common in children treated for ALL and AML, how are increases in the ferritin level due to infections handled in the study?
2) One of the major prognostic factors in ALL and AML are the presence of major treatment-stratifying genetic alterations. Patients with high-risk genetic alterations, e.g., KMT2A-rearranged, hypodiploidy, receive more intensive treatment and are thus more likely to suffer from treatment-induced toxicity, which includes anemia. Subsequently, these children may receive more blood transfusions. Furthermore, certain medications used are more likely to cause anemia, and toxicity during the treatment may be the reason for treatment adaptations. Authors should consider using risk-stratifying alterations in their model and correlate ferritin levels with the presence of these genetic alterations at diagnosis.
3) It is unclear why ferritin levels in children with AML show opposing trend when it comes to outcome compared to children with ALL. Authors should explain this in more detail.
4) Discussion is very long and I believe that condensing the text and streamlining it to the authors main findings would be improve the readability and bring forward the main conclusions.
5) The absolute number of patients at risk at each timepoint should be added below the survival curves. Some of the curves seem to represent only a few patients, and adding this information will aid in interpreting the data.
Minor comments:
1) In the table 1, column “Leukemias” what is the number showing min-max for males and females? Was this supposed to be the range as for the Total?
2) Table 2 is rather complex and essentially shows two different groups of information. Thus, I would advise to split this table in two, one showing age groups and the other showing treatment.
Comments on the Quality of English LanguageQuality of English language is high.
Author Response
Thank you very much for giving me the opportunity to submit a revised draft of my manuscript titled ,,Iron overload in children with acute leukemias – experience of one center’’ to Cancers. I appreciate the time and effort that you have dedicated to providing your valuable feedback on my manuscript. I am grateful for all comments on my work. I have been able to incorporate changes to reflect most of the suggestions provided by your review.
In answer to comments :
Comment 1: Authors do not show data about infections and inflammations during the treatment. Since these are quite common in children treated for ALL and AML, how are increases in the ferritin level due to infections handled in the study?
CRP and PCT were evaluated at the beginning of the treatment, correlations between these parameters and ferritin concentration were shown in results(section 3.3). During treatment we didn’t perform that type of analysis, because ferritin concentration was not evaluated during inflammatory process and infections (we added inclusion criteria to the section 2.1). We performed analysis of ferritin concentration and inflammatory parameters to exclude only inflammatory background of initially elevated ferritin concentration in our patients, but we removed this section from the manuscript, because of its length. These are results not showed in the manuscript:
Ferritin concentration and inflammatory parameters
In order to exclude inflammatory influence on ferritin concentration we examined initial concentration of C-reactive protein (CRP) and procalcitonin (PCT) in total group and in groups according to specific diagnoses. Median initial concentration of CRP in total group was 3.7 mg/l (95% Conf. Interval 2.331-5.116)and median initial concentration of PCT was 0.18 ng/ml(95% Conf. Interval 0.124-0.210. In ALL patients median initial CRP concentration was 3.78 mg/l ((95% Conf. Interval 2.349-5.290) and median PCT concentration was 0.15 ng/ml (95% Conf. Interval 0.1-0.21) respectively. In AML group median initial CRP concentration was 2.35 mg/l ((95% Conf. Interval 0.716-68.710) and median PCT concentration was 0.21 ng/ml (95% Conf. Interval 0.12-4.15). In total group we did not find statistically important correlations between CRP and ferritin concentration nor between PCT and ferritin (r=-0,082, p=0.938; r=0.056, =0.74). Analogous analysis performed in ALL and AML groups didn’t show correlations between inflammatory parameters and ferritin concentrations in the moment of diagnosis ferritin and CRP( ALL: ferritin and CRP r=-0.012, p=0.90, ferritin and PCT r=0.07, p=0.7021; AML: ferritin and CRP r=0429, p=0.33, ferritin and PCT r=0.132, p=0.83).
Subsequent measurements of ferritin concentration during the treatment (SF1-SF5 and SF-finish) were performed under the requirement of absence of inflammation.
Upon this analysis we excluded only inflammatory background of initially elevated ferritin concentration in our patients diagnosed with leukemia.
Comment 2: One of the major prognostic factors in ALL and AML are the presence of major treatment-stratifying genetic alterations. Patients with high-risk genetic alterations, e.g., KMT2A-rearranged, hypodiploidy, receive more intensive treatment and are thus more likely to suffer from treatment-induced toxicity, which includes anemia. Subsequently, these children may receive more blood transfusions. Furthermore, certain medications used are more likely to cause anemia, and toxicity during the treatment may be the reason for treatment adaptations. Authors should consider using risk-stratifying alterations in their model and correlate ferritin levels with the presence of these genetic alterations at diagnosis.
Thank you very much for this important comment. We agree that analysis including genetic alterations affecting the outcome of patients with acute leukemias. We stratified our patients into risk groups, which also include on genetic alterations. High risk groups require more transfusions and in some cases patients undergo alloHSCT, which also increases significantly the number of transfusions. In order to clarify our analyzed group we did not detailed cytogenetic parameters in the manuscript. I hope that in subsequent analyses, based on larger group of patients we will be able to perform objective analysis including genetic alterations.
Comment 3:v It is unclear why ferritin levels in children with AML show opposing trend when it comes to outcome compared to children with ALL. Authors should explain this in more detail.
We explained this in discussion part by adding the citation of Ihlow et al, whose analysis showed opposite results. It iis possible the low numer of patients could influence the obtained results and should be analyzed on bigger group of patients. The difference in the outcomes between adults and children could also influence this discrepancy.
,,No statistically significant differences were demonstrated in context of transfused blood units’ levels and survival. However, AML patients with more units implemented seemed to survive longer than their counterparts with less blood used, but the result not achieved statistical significance. Our results are opposite to those obtained by Ihlow et al. who showed that high SF has a negative impact on long-term survival in 137 adult patients with AML. We explain this discrepancy by the small number of subjects and possible differences in outcomes between children and adults.
Comment 4: Discussion is very long and I believe that condensing the text and streamlining it to the authors main findings would be improve the readability and bring forward the main conclusions.
Thank you very much, we realize that the discussion section is quite long. However, in our opinion, shortening the discussion would influence the opportunity to thoroughly explain the results obtained. If it is necessary to shorten this part of manuscript, we are ready to follow your suggestions.
Comment 5: The absolute number of patients at risk at each timepoint should be added below the survival curves. Some of the curves seem to represent only a few patients, and adding this information will aid in interpreting the data.
The absolute number of analyzed patients for each survival curve was given in text and tables. Additional table will affect the clarity of those figures.
Minor comments:
Comment 1: In the table 1, column “Leukemias” what is the number showing min-max for males and females? Was this supposed to be the range as for the Total?
Thank you very much for taking close look at the table. We have corrected this error.
Comment 2: Table 2 is rather complex and essentially shows two different groups of information. Thus, I would advise to split this table in two, one showing age groups and the other showing treatment.
Thank you for this comment. Indeed, table 2 seem difficult to read. We decided to divide this table into 2 smaller in order to make them clear.
Thank you one more time for your valuable comments,
Kind regards,
Malgorzata Sawicka-Zukowska
Reviewer 2 Report
Comments and Suggestions for Authors
Dear Authors,
Thank you very mich for submitting your paper in the Cancers.
- In the title, you can remove '' as well as a dot at the end. In the title you could also indicate which center specifically or the country in which the study was undertaken. You could also specify which acute leukemias are you considering - the title would be more informative
- line 19 correct to 'however, THIS can...'
- in the abstract, please indicate the mean age of the studied population
- please provide full names of the abbreviations before using them - e.g. AML and ALL, SF, etc.
- Introduction, lines 36-41 are not needed, I would recommend removing this part
- As you included only those patients with numerous transfusions, what was the control group to which you compared your results?
- Line 99, what type of leukemia?
- Thank you for the exclusion criteria, but you should also include inclusion criteria
- Regarding the variables that were collected from all the patients you are mentioning - did you also collect the data regarding the medications taken by the patients? It is crucial since various therapeutics might affect serum levels of various micro/macro-nutrients
- Table 1 is quite blurred, please correct it
- what does n=30 in Table 2 mean?
- in table 2 all the abbreviations should be mentioned in the legend below the table. Further, the table is not clear and should be reorganized
- Line 112, serum ferritin - abbreviation SF was mentioned in the text earlier without explanations. Please correct
- Table 4 - please unify it so that it is stylistically correct
- Regarding inclusion/exclusoin criteria - did you consider any special diets/supplementation intake by the children?
Comments on the Quality of English LanguageThe English in the manuscript requires extensive corrections, there are numerous grammatical errors that should be corrected. I recommend extensive editing or having the paper checked by the native speaker.
Author Response
Thank you very much for giving me the opportunity to submit a revised draft of my manuscript titled ,,Iron overload in children with acute leukemias – experience of one center’’ to Cancers. I appreciate the time and effort that you have dedicated to providing your valuable feedback on my manuscript. I am grateful for all comments on my work. I have been able to incorporate changes to reflect most of the suggestions provided by your review.
In answer to comments :
Comment 1: ,,In the title, you can remove '' as well as a dot at the end. In the title you could also indicate which center specifically or the country in which the study was undertaken. You could also specify which acute leukemias are you considering - the title would be more informative
- I removed the ,, ‘’ symbol.
- I changed the title for: Iron overload in children with acute lymphoblastic and acute myeloblastic leukemia – experience of one center
Comment 2: line 19 correct to 'however, THIS can...'
- I corrected the sentence for: However, this can lead to post transfusion iron overload.
Comment 3: in the abstract, please indicate the mean age of the studied population
We performed longitudinal study of PRBC transfusions and changes in ferritin concentrations during oncological treatment of 135 patients with childhood acute lymphoblastic and acute myeloblastic leukemia (ALL and AML, median age 5,62 years).
Comment 4. please provide full names of the abbreviations before using them - e.g. AML and ALL, SF, etc.
I provided full names of the abbreviations in the text before using them.
Comment 5. Introduction, lines 36-41 are not needed, I would recommend removing this part
Comment 6. As you included only those patients with numerous transfusions, what was the control group to which you compared your results?
Analyzed group composed of pediatric oncological patients with the history of numerous transfusions. It would be very difficult to find controls with the same medical burden as those analyzed. That is why we decided to base our analysis on laboratory norms adjusted for age and gender. Comparing SF and number of transfusion between leukemic patients and patients with solid tumors in our other analysis (in wasn’t mentioned in actual manuscript) we observed statistically significant differences in ferritin concentrations at time points and in number of transfusions. However, taking under consideration the actual size of the manuscript we decided to use laboratory norms as reference.
Comment 7. Line 99, what type of leukemia?
I am not sure if the line number is correct.
Comment 8. Thank you for the exclusion criteria, but you should also include inclusion criteria
I added the Inclusion criteria:
- age 0-18 years
- diagnosis with acute lymphoblastic leukemia or acute myeloblastic leukemia
- history of blood transfusions during antineoplastic treatment
- no previous history of multiple transfusions preceding diagnosis of leukemia
- ferritin concentration obtained in numerous time points during treatment
Comment 9. Regarding the variables that were collected from all the patients you are mentioning - did you also collect the data regarding the medications taken by the patients? It is crucial since various therapeutics might affect serum levels of various micro/macro-nutrients
In pediatric population comparing to adults we observe rare cases of previous medication history. None of analyzed patients was treated with oral or parenteral iron at least 1 year before the diagnosis of leukemia. It is not recommended taking iron during antineoplastic treatment due to impaired iron metabolism and post-transfusion iron burden. None of analyzed patients underwent previous the treatment with medications, which could influence on SF.
Comment 10. Table 1 is quite blurred, please correct it
|
|
LEUKEMIAS |
ALL* |
AML** |
|||||||||||||
|
TOTAL |
MALE |
FEMALE |
TOTAL |
M |
F |
Total |
M |
F |
||||||||
|
Number |
135 |
77 |
58 |
110 |
64 |
46 |
35 |
23 |
12 |
|||||||
|
% |
100 |
57.03 |
42.97 |
100 |
58.2 |
41.82 |
100 |
65.71 |
34.28 |
|||||||
|
Median Min-max |
|
|
|
|
||||||||||||
*ALL Acute lymphoblastic leukemia
**AML Acute myeloblastic leukemia
I corrected table 1.
Comment 11. what does n=30 in Table 2 means?
N=30 in table 1 means total number of patients who underwent the procedure of alloHSCT
Comment 12. in table 2 all the abbreviations should be mentioned in the legend below the table.
I added the abbreviations in the legend below the table.
Comment 13: Line 120:
I changed
Serum ferritin (SF) was used as a marker of iron overload (IO). Serum ferritin concentration (SF, ng/ml) was measured using electrochemiluminescence method in every patient.
For
Serum ferritin (SF, ng/ml) was used as a marker of iron overload (IO). SF was measured using electrochemiluminescence method in every patient.
Comment 14. Table 4 - please unify it so that it is stylistically correct
In table 4 we present patients with SF elevated above 500 ng/ml and 1000 ng/ml at the begging of the treatment and at the cessation of the treatment. Last column named: ,,same patient number’’, shows number of patients, who started treatment with elevated SF and finished treatment with elevated SF. The percentage was calculated from the total number of patients, who started treatment with elevated SF in particular group. For example: 53 of 55 patients (96,35%) who started treatment with SF concentration above 500 ng/ml finished treatment with SF above 500 ng/ml. 19 patients started treatment with SF above 1000 ng/ml and 14 patients from these 19 finished treatment with SF >1000 ng/ml. The table shows the tendency for increased SF at the finish of the therapy in patients with leukemias who started with the increased SF.
Comment 15. Regarding inclusion/exclusoin criteria - did you consider any special diets/supplementation intake by the children?
None of analyzed patients received vegetarian or vegan diet before treatment, as well as none of them received iron or vitamin B6, B12 in therapeutic doses before the oncological therapy. It is not recommended for children during oncological treatment to take iron supplementation during therapy.
Thank you one more time for the review, I can provide more answers if needed.
Sincerely,
Malgorzata Sawicka-Zukowska
Reviewer 3 Report
Comments and Suggestions for Authors
In their manuscript, Sawicka-Zukowska et al. share clinical observations and analyses to evaluate the role of ferritin as a non-specific marker of leukemic growth and marker of transfusion-related iron overload. I would like to thank the authors for sharing their observations on this important clinical issue.
The main clinical parameters considered for multivariate analysis, as well as the criteria for statistical analysis are adequate. As a major overall comment, please note that it is misguiding to the reader that “treatment response” is used to describe survival/death risk. There is no information on blast percentage or other remission-defining parameters shared in this study. Please edit to specify the correct parameters evaluated in the study where appropriate.
Of note, the authors describe that low serum ferritin concentrations may predict for poor prognosis in AML, which is contrary to similar observations in the literature in different patient cohorts (for example, doi: 10.5603/AHP.2021.0008, 10.1080/10428194.2018.1461860). This manuscript can be improved if more information pertaining disease severity is provided to the readers. If the information on risk stratification, disease relapse and treatment regimens is available, it would greatly enhance this manuscript.
Major comments:
- The chemotherapy treatment regimens for the patients must be specified. Studies have demonstrated associations with treatment intensity in AML patients (Ihlow et al., 2019; doi: /10.1080/10428194.2018.1461860). Given the findings on the correlation between increased survival in AML patients and low SF levels, it would greatly enhance the study to assess possible confounding variables. If the type of treatment was the same for all patients, this must be specified.
- If the information is available, disease risk classification (ELN) would also be a useful factor for multivariate analysis.
- If the information is available, please specify which cases were relapsed leukemias.
Minor corrections:
- In the simple summary, please specify that iron overload is a side effect of cancer therapies given the requirement for blood transfusions.
- In the simple summary, line 14, please consider that response to treatment is not specified in terms of remission, which the reader may assume from reading the summary. The statement “We showed that serum ferritin levels could be a valuable tool in monitoring the response to oncological treatment in children with AML and ALL” can imply that each round of therapy was evaluated against transfusion requirements and iron overload, which is not part of the study design. Please correct this statement to read that SF may be used as a prognostic factor for death risk, as it is shown in the figures.
- In line 26, begging should spell “beginning”.
- The figure and table panels have inconsistent fonts and spacing. Please consider generating them again to display consistent font sizing.
- At the end of the introduction, please re-state the aim of the study to guide the reader.
- In line 69, please specify whether the transfusion units per month are for the treatment of non-malignant hematological disorders.
- In lines 87-88, please specify whether the assessment of iron overload in childhood cancer is part of the standard of care.
- The discussion can be substantially condensed. Please consider thorough editing for conciseness.
Comments on the Quality of English LanguageThe manuscript is understandable and detailed as it is presented. I would like to thank the authors for providing extensive information and context to the reader in the discussion section.
I would recommend minor English grammatical editing to correct a number of instances where definite and indefinite articles are misused.
In addition, the manuscript is very long in terms of length and flow. The discussion section can be condensed substantially without missing the key information and context. Please consider substantial editing to improve the conciseness of the text throughout the entire manuscript.
Author Response
Thank you very much for giving me the opportunity to submit a revised draft of my manuscript titled ,,Iron overload in children with acute leukemias – experience of one center’’ to Cancers. I appreciate the time and effort that you have dedicated to providing your valuable feedback on my manuscript. I am grateful for all comments on our work. I have been able to incorporate changes to reflect most of the suggestions provided by your review.
In answer to comments :
Major comments:
Comment 1: The chemotherapy treatment regimens for the patients must be specified. Studies have demonstrated associations with treatment intensity in AML patients (Ihlow et al., 2019; doi:/10.1080/10428194.2018.1461860). Given the findings on the correlation between increased survival in AML patients and low SF levels, it would greatly enhance the study to assess possible confounding variables. If the type of treatment was the same for all patients, this must be specified.
We explained this in discussion part by adding the citation of Ihlow et al, whose analysis showed opposite results. It is possible the low number of patients could influence the obtained results and should be analyzed on bigger group of patients. The difference in the outcomes between adults and children could also influence this discrepancy.
,,No statistically significant differences were demonstrated in context of transfused blood units’ levels and survival. However, AML patients with more units implemented seemed to survive longer than their counterparts with less blood used, but the result not achieved statistical significance. Our results are opposite to those obtained by Ihlow et al. who showed that high SF has a negative impact on long-term survival in 137 adult patients with AML. We explain this discrepancy by the small number of subjects and possible differences in outcomes between children and adults.
Comment 2: If the information is available, disease risk classification (ELN) would also be a useful factor for multivariate analysis.
This data were not available, in our pediatric practice we don’t use ELN system commonly. Comment 3: If the information is available, please specify which cases were relapsed leukemias.
General information about relapse patients are shown in table 2: 4 with AML (4 from High risk group, 4 wit ALL – 2 IRG, 2 HRG)
Minor corrections:
Comment 1: In the simple summary, please specify that iron overload is a side effect of cancer therapies given the requirement for blood transfusions.
We changed sentence for: ,,Post-transfusion iron overload is a common side effect of anticancer treatment, including leukemias.’’
Comment 2: In the simple summary, line 14, please consider that response to treatment is not specified in terms of remission, which the reader may assume from reading the summary. The statement “We showed that serum ferritin levels could be a valuable tool in monitoring the response to oncological treatment in children with AML and ALL” can imply that each round of therapy was evaluated against transfusion requirements and iron overload, which is not part of the study design. Please correct this statement to read that SF may be used as a prognostic factor for death risk, as it is shown in the figures.
We changed sentence for: ,,Post-transfiusion iron overload is a common side effect of anticancer treatment, including leukemias. We showed that serum ferritin levels could be valuable prognostic marker of death in children with AML and ALL.’’
Comment 3: In line 26, begging should spell “beginning”.
We corrected the error.
Comment 4: The figure and table panels have inconsistent fonts and spacing. Please consider generating them again to display consistent font sizing.
Thank you for this comment, we reconstructed the tables.
Comment 5 : At the end of the introduction, please re-state the aim of the study to guide the reader.
We added text at the end f the introduction: The aim of the study was to evaluate the changes in ferritin concentration due to transfusion during antineoplastic treatment. In addition we analyzed prognostic value of iron levels in assessment of death risk in children with acute leukemias.
Comment 6: In line 69, please specify whether the transfusion units per month are for the treatment of non-malignant hematological disorders.
We added: It was also shown that the frequency of transfusion in the amount of 2 units per month used in non-malignant hematological disorders will result in the accumulation of 20 g of extra iron in the body within 4 years.
Comment 7: In lines 87-88, please specify whether the assessment of iron overload in childhood cancer is part of the standard of care.
We changed sentence for: ,,The most common side effect, connected with multiple blood transfusions, is iron overload (IO) rutinly evaluated after procedures.’’
Coomment 8: The discussion can be substantially condensed. Please consider thorough editing for conciseness.
Thank you very much, we realize that the discussion section is quite long. However, in our opinion, shortening the discussion would influence the opportunity to thoroughly explain the results obtained. If it is necessary to shorten this part of manuscript, we are ready to follow your suggestions.
Thank you one more time for your valuable comments,
Kind regards,
Malgorzata Sawicka-Zukowska
Round 2
Reviewer 1 Report
Comments and Suggestions for Authors
Author have answered to all of my comments and adapted manuscript accordingly.
I still believe manuscript will benefit from focusing discussion on important points. I also believe that showing data related to genetic subtype of leukemia (high-risk in particular), at least in the supplement, would be of added value to the manuscript.
Comments on the Quality of English LanguageOverall quality of English language is good. Minor grammatical errors and spelling mistakes are present.
Author Response
Thank you very much for giving us the opportunity to submit the improved version of our manuscript. As suggested, we shortened the discussion to focus on the most important points. We added an appendix containing molecular cytogenetics subtypes in high-risk patients with acute lymphoblastic leukemia (table) and a graph showing differences between HR and non-HR patients in terms of ferritin concentrations, blood transfused per kg and units of blood blood transfused.
Thank you for the time and your effort that you have dedicated to providing your valuable feedback on our manuscript.
Kind regards,
Malgorzata Sawicka-Zukowska
